# Antioxidant and *In Vitro* Hepatoprotective Activities of a Polyphenol-Rich Fraction from the Peel of *Citrus lumia* Risso (Rutaceae)

**DOI:** 10.3390/plants14081209

**Published:** 2025-04-15

**Authors:** Vincenzo Musolino, Antonio Cardamone, Rosario Mare, Anna Rita Coppoletta, Francesca Lorenzo, Francesca Rita Noto, Angelo Galluccio, Luigi Tucci, Carmine Lupia, Cristina Carresi, Mariangela Marrelli, Samantha Maurotti, Micaela Gliozzi, Tiziana Montalcini, Arturo Pujia, Vincenzo Mollace

**Affiliations:** 1Laboratory of Pharmaceutical Biology, Department of Health Sciences, Institute of Research for Food Safety & Health IRC-FSH, University “Magna Græcia” of Catanzaro, 88100 Catanzaro, Italy; 2Laboratory of Pharmacology, Department of Health Sciences, Institute of Research for Food Safety and Health IRC-FSH, University Magna Græcia of Catanzaro, 88100 Catanzaro, Italy; tony.c@outlook.it (A.C.); annarita.coppoletta1@gmail.com (A.R.C.); carresi@unicz.it (C.C.); mollace@unicz.it (V.M.); 3Department of Medical and Surgical Sciences, University “Magna Græcia” of Catanzaro, 88100 Catanzaro, Italy; mare@unicz.it (R.M.); francescarita.noto@studenti.unicz.it (F.R.N.); pujia@unicz.it (A.P.); 4Department of Health Science, AGreen Food Research Center, University Magna Græcia of Catanzaro, 88100 Catanzaro, Italy; francesca.lorenzo@studenti.unicz.it; 5Department of Clinical and Experimental Medicine, University Magna Graecia, 88100 Catanzaro, Italy; angelo.galluccio@studenti.unicz.it (A.G.); smaurotti@unicz.it (S.M.); tmontalcini@unicz.it (T.M.); 6H&AD Srl, 89032 Bianco, Italy; l.tucci@head-sa.com; 7Mediterranean Ethnobotanical Conservatory, 88054 Sersale, Italy; studiolupiacarmine@libero.it; 8National Ethnobotanical Conservatory, 85040 Castelluccio Superiore, Italy; 9Department of Pharmacy, Health and Nutritional Sciences, University of Calabria, 87036 Rende, Italy; mariangela.marrelli@unical.it; 10Research Center for the Prevention and Treatment of Metabolic Diseases, University “Magna Græcia”, 88100 Catanzaro, Italy

**Keywords:** *Citrus lumia* Risso, electron paramagnetic resonance (EPR) spectroscopy, antioxidant activity, metabolic dysfunction–associated steatotic liver disease (MASLD), spheroids, lipid accumulation, polyphenols

## Abstract

*Citrus lumia* Risso is an ancient, cultivated Mediterranean lime belonging to the Rutaceae family. It is a species extremely difficult to retrieve, but it is still found in some private gardens in certain regions of Southern Italy. *Citrus* fruits are a rich source of bioactive compounds, particularly polyphenols, which have been linked to a reduction in the risk of several metabolic diseases. Here, hesperidium peel extracts were obtained by maceration with ethanol:water mixtures in different proportions (50:50, 80:20, 0:100) and the resulting crude extracts were then passed through a glass column containing adsorbent resins to concentrate the polyphenolic compounds. After phytochemical characterization, the extracts were evaluated for antioxidant activity using electron paramagnetic resonance (EPR) spectroscopy. Finally, the water polyphenolic-rich extract (ClumWp), which was the extract with the highest flavonoid content (18.355 ± 1.607 mg/mL) and the strongest antioxidant activity against hydroxyl radical, was tested to evaluate its potential protective effects on lipid accumulation in both 2D hepatocyte cultures and 3D spheroids. Treatment with 25 and 50 μg/mL resulted in a reduction in intracellular lipid content in the HepG2 liver cell line, while treatment with 100 µg/mL ClumWp resulted in a reduction in the intracellular lipid content in HepG2 + LX2 spheroids. In addition, treatment with ClumWp significantly increased ATP levels in the spheroids compared to those untreated, suggesting its ability to restore and promote ATP production. Our results highlight that the study of neglected species, such as *Citrus lumia* Risso, remains a valuable opportunity to valorize Mediterranean biodiversity, especially in the context of its potential applications to improve human health. In particular, the polyphenolic fraction of *Citrus lumia* peel showed promising effects on lipid metabolism and cellular energy balance and may prove valuable in the treatment of metabolic disorders such as MASLD, where lipid accumulation disrupts normal cellular functions.

## 1. Introduction

*Citrus* L. is a genus belonging to the Rutaceae family. Though it represents a major fruit crop worldwide, global production is declining due to reduced production caused by unfavorable weather and yield [1]. *Citrus* genus has a wide distribution, covering many parts of the world, and is particularly well known in tropical and subtropical regions [2].

Among the various species, *Citrus lumia* Risso is an old lime that has been cultivated in the Mediterranean basin since ancient times. Currently, it is extremely difficult to retrieve, but it is still present, albeit rarely, in some private gardens in certain regions of southern Italy, especially in Campania, Sicily, and Calabria [3]. 

The binomial *Citrus lumia* was formally assigned by Risso in 1826 [4], and the species has been described by several systematists of the genus *Citrus*, who confirmed the hybrid nature of *Citrus lumia* Risso [3]. Indeed, *Citrus lumia* Risso has been recognized as a hybrid of citron and orange (*C. medica* × *C. sinensis*), or of citron and pummelo (*C. medica* L. × *C. grandis*), or, again, of pummelo and lemon (*C. grandis* × *C.* × *limon*) [3]. According to the online taxonomic database Plants of the World Online (POWO), published by the Royal Botanic Gardens, Kew, *Citrus* × *lumia* Risso is an artificial hybrid, and the hybrid formula of this artificial cross is *C. maxima* × *C. medica* [5]. Nevertheless, there is still much to be studied and discovered about its identity, as evidenced by World Flora Online, which lists *Citrus lumia* Risso as a synonym of *Citrus medica* L. [6].

It is a small tree, and the leaves are obovate with serrate margins and sometimes slightly winged petioles. The scented flowers have white petals that are externally reddish. The hesperidium is globose, about 6 cm in diameter, with a depressed base and apex. Morphologically, the base is very peculiar, characterized by a flattened umbo and a prominent groove that surrounds it. In the ripe fruit, the smooth and aromatic exocarp (flavedo) is yellow-sulphureous in color. The mesocarp (albedo) is white, spongy, and it has a bitter taste. The membranous endocarp consists of 9–11 segments within which vesicles rich in delicate, non-acid, and sweet juice develop (Figure 1).

*Citrus* fruits are particularly appreciated for their beneficial effects on human health, as they are a relevant source of bioactive compounds with high antioxidant properties, such as ascorbic acid and flavonoids [7]. *Citrus lumia* Risso is known in Sicily as *lumia* [3], in the Neapolitan area as *limmo, limo*, or *limma* [8], and in some areas of Calabria as *limuncegliu* among local farmers (personal ethnobotanical interview) or more commonly as *lemoncetta Locrese* [8]. Its juice is characterized by the presence of four main flavonoids: eriocitrin, hesperidin, diosmin, and rutin [8]. On the other hand, the mesocarp of the fruit reveals the presence of flavones (i.e., eriocitrin, hesperidin, tangeretin, and diosmin), phenolic acids (i.e., chlorogenic acid and ferulic acid), and flavonols (i.e., hyperoside and rutin) [9]. Among these, eriocitrin and hesperidin represent two compounds of interest due to their capability to efficaciously counteract oxidative stress in metabolic disorders and in other chronic diseases [10,11]. Oxidative stress, as well as lipid accumulation in the liver cells, represents the initial events responsible for the onset of Metabolic Dysfunction-Associated Steatotic Liver Disease (MASLD), the most common liver disease worldwide, affecting more than 30% of the global population [12]. Given the high incidence, early intervention to prevent the consequences of the imbalanced energy status, free radical overproduction, and metabolic alteration in hepatocytes remains a priority [13]. Mitochondria are the main organelles responsible for the synthesis of adenosine triphosphate (ATP) through the electron transport chain. This reaction is characterized by the formation of reactive oxygen species (ROS), especially superoxide anion, as end products that, in turn, are neutralized by endogenous defenses, such as mitochondrial superoxide dismutase [14]. As carbohydrate and fatty acid oxidation pathways mainly occur in mitochondria, they play a pivotal role in the energy metabolism of the cell [15]. In response to stress stimuli, such as over-nutrition in obese patients [16], energy and oxidative metabolism can be severely imbalanced, resulting in oxidative stress, impaired lipid metabolism, and activation of apoptotic cell death [14]. In this scenario, the identification of bioactive compounds from *Citrus lumia*, capable of maintaining mitochondrial homeostasis and counteracting oxidative stress induced by several metabolic disorders, can add novel value to the knowledge of *Citrus* species of the Mediterranean basin. Therefore, the aim of our study was to perform a phytochemical analysis of *Citrus lumia* Risso peel extracts both before and after a process to concentrate the polyphenolic fraction and increase the yield of bioactive compounds. We evaluated the antioxidant capacity of these polyphenol-rich extracts using electron paramagnetic resonance (EPR) spectroscopy. While previous findings have shown that *Citrus lumia* Risso exerts antioxidant, anti-inflammatory and anti-angiogenic properties, no studies have investigated its potential effects on lipid accumulation. To address this gap, our study also focused on evaluating the protective effects of polyphenol-rich *Citrus lumia* peel extracts on lipid accumulation in both 2D hepatocyte cultures and spheroids.

## 2. Materials and Methods

### 2.1. Plant Material

Fresh fruits of *Citrus lumia* Risso were manually harvested in April 2024 by a local farmer in Benestare (località Drafà, 38°10′14′′ N 16°08′16′′ E), a small town in the province of Reggio Calabria (Italy). The area is characterized by typical farmland of the Mediterranean scrub land, with a mild climate and a humidity level of 68% at the time of harvest, which contributes to the optimal growing conditions for this *Citrus* variety. The taxonomic identification has been confirmed by Dr. C. Lupia. Voucher specimens have been deposited at the Mediterranean Ethnobotanical Conservatory (Sersale, Catanzaro, Italy) under the following accession numbers: *Citrus lumia* Risso: Rutaceae section, 18.

### 2.2. Extraction Procedure

After harvesting, the hesperidia were transported to the laboratory of Pharmaceutical Biology, Department of Health Sciences, Institute of Research for Food Safety and Health IRC-FSH, University “Magna Græcia” of Catanzaro, for processing. First, the fruits were thoroughly washed with distilled water to remove impurities from the epicarp surface. The peel was dried at 40 °C for 24 h and then ground to a fine powder in liquid nitrogen using a mortar and pestle with a high extraction surface. Then, 10 grams of minced peel were transferred to a 200 mL glass flask and macerated at 40 °C, in the dark, with 150 mL of solvent for 24 h. The solvent was a mixture of ethanol:water (EtOH:w, % *v*/*v*) in different proportions: EtOH/w, 50/50, EtOH/w 80/20, and EtOH/w 0/100. After the extraction, the plant matrix was removed by centrifugation at +4 °C, 1036 RCF for 10 min and subsequently filtered through a filter paper. The hydroalcoholic extracts (Et50 and Et80) were first concentrated at 45 °C using a rotary evaporator (DLAB—RE100S) coupled to a N820 LABOPORT pump; then the samples were frozen at −80 °C for 24 h and then left at RT until the liquid state were reached again. The samples were then frozen with liquid nitrogen and lyophilized at −70 °C for 48 h using a lyophilizator (LIO-5PDGT VC, 5 Pascal, Italy) to obtain crude extracts: 2.46 g and 1.48 g, respectively. The water extract was frozen at −80 °C for 24 h, allowed to reach the liquid state at room temperature, frozen with liquid nitrogen, and lyophilized at −70 °C for 48 h to give 6.3 g of crude extract (W100%).

### 2.3. Preparation of Citrus lumia Extracts Concentrated in Polyphenols

The crude extracts were diluted with distilled water to a total volume of 200 mL and then passed through a glass column containing 100 mL of adsorbent resins (Sepabeads SP 700, Mitsubishi Chemical Corp, 40549 Düsseldorf, Germany) to concentrate the polyphenolic compounds [17]. The column was washed with distilled water to remove impurities and then eluted with 250 mL of 50% ethanol. The ethanol eluate from each extract was collected and re-concentrated using a rotary evaporator under vacuum, yielding 0.605 g of dry extract from W100 (ClumWp, 6.05% yield); 0.980 g from Et50 (ClumEt50p, 9.8% yield) and 0.730 g from Et80 extract (ClumEt80p, 7.3% yield).

### 2.4. High Performance Liquid Chromatography (HPLC) Analysis

Eriocitrin and hesperidin were quantified by High-Performance Liquid Chromatography coupled to Diode-Array Detection (HPLC-DAD) [17]. Reference standards were purchased from Sigma-Aldrich (>99% purity). The HPLC-DAD analysis was performed on a Perkin-Elmer Flexar Module equipped with a C18 column (250 mm × 4.6 mm, 5 µm particle size). The mobile phase consisted of a gradient of 0.1% trifluoroacetic acid (A) and acetonitrile (B), with A going from 85% to 78% in 8 min, maintaining 78% for 4 min, then to 40% in 4 min, to 0% in 1 min. The flow rate was set at 1.1 mL/min, and the detection wavelength was set at 284 nm. The injection volume was 3 µL, and the column temperature was maintained at 30 °C. The retention times for eriocitrin and hesperidin were determined to be 8.3 min and 12 min, respectively. The concentration of these compounds in each extract was quantified based on calibration curves established with standard solutions in the range of 10–1000 ppm. The calibration curves showed excellent linearity (R^2^ > 0.999) over the concentration range used.

### 2.5. Total Flavonoid Content

Total flavonoid content (TFC) was measured using a recent UV-Vis spectrophotometric method [18]. Briefly, 1 mL of the sample was incubated with aqueous sodium nitrite solution (60 µL—5% *w*/*v*), vortexed, and allowed to react for 5 min. Following this, 120 µL of 10% (*w*/*v*) aqueous aluminum chloride solution was added, and the mixture was vortexed and incubated again. The pH was then adjusted to neutral by the addition of 0.40 mL of 1 M sodium hydroxide. The final sample volume was 2 mL. Absorbance was recorded immediately at the peak wavelength (λ~341 nm). For analytical measurements, a calibration curve was established using a mixture of naringin, apigenin, rutin, and hesperidin in an equal weight ratio.

### 2.6. Evaluation of Radicals Scavenging by Electron Paramagnetic Spectroscopy (EPR)

Antioxidant activity was investigated using Electron Paramagnetic Resonance (EPR) spectroscopy, following previously described protocols [19,20].

For the DPPH (2,2-diphenyl-1-picrylhydrazyl) radical scavenging assay, 50 μL of *Citrus lumia* extract (5 mg/mL) was combined with 200 μL of a 1 mM methanolic DPPH solution (D9132-5G, Sigma-Aldrich, St. Louis, MO, USA). The reaction mixture was incubated in the dark for 1 min and EPR spectra were recorded at room temperature using a Bruker Magnettech ESR5000 (Bruker Biospin MRI GmbH, Ettlingen, Germany).

The spin trapping technique was used to determine the scavenging activity against hydroxyl radicals (OH^•^). This technique, which stabilizes reactive oxygen species (ROS) characterized by a very short half-life, involves the reaction of a nitrone or a nitrous compound with the free radical to form a more stable adduct. BMPO (5-tert-butoxycarbonyl-5-methyl-1-pyrroline-N-oxide) was used to form the stable adduct with OH^•^ radicals, generated by the Fenton reaction. The reaction mixture contained 15 μL of the BMPO (B568-10, Dojindo EU GmbH, Munich, Germany), 75 μL of 1 mM H_2_O_2_, 75 μL of 100 μM iron (II) sulphate heptahydrate (FeSO_4_•7H_2_O, 7782-63-0, Merck, Darmstadt, Germany), and 50 μL of ddH_2_O. Following the adduct formation, 50 μL of the *Citrus lumia* extract was added, and the mixture was incubated for 1 min before acquiring the EPR spectra.

The instrument was set to the following parameters: 9.43 GHz X-band, 0.2 mT (DPPH assay) or 0.05 mT (OH^•^ assay) modulation amplitude, 336.64 (DPPH assay) or 335.46 (OH^•^ assay) mT central field, 12.00 mT sweep, 30 s sweep time, 100 Khz modulation frequency, 20 mW (DPPH assay) 6 mW (OH^•^ assay) microwave power, and 3 accumulations.

The spectral regions were analyzed using the ESRStudio software (v. 1.74.6, Bruker Biospin, Ettlingen, Germany), and the scavenging activity was quantified based on the reduction in EPR radical signal intensity upon the addition of the *Citrus lumia* extracts. The scavenging percentage was determined using the formula: scavenger % = (A_0_ − A_extract_/A_0_) × 100, where A_0_ represents the control signal intensity and A_extract_ corresponds to the signal intensity after extract addition.

Ascorbic acid (5 mg/mL), a well-established antioxidant, was used as the positive control in both EPR analyses.

### 2.7. 2D and 3D Cell Culture

HepG2 hepatocellular carcinoma cells were obtained from the American Type Culture Collection (ATCC) and cultured in MEM (Sigma Aldrich, St. Louis, MI, USA) supplemented with 10% FBS and 1% penicillin-streptomycin (PAA, Linz, Austria). Cells were maintained at 37 °C in a 5% CO_2_ atmosphere and sub-cultured twice weekly after trypsinization. The HepG2 + LX-2 spheroids were prepared as previously described [21]. Briefly, HepG2 cells and LX-2 cells (ATCC) were combined in a 24:1 ratio and seeded into 96-well round-bottomed ultra-low attachment plates (Corning) at a density of 2000 cells/well. The cells were cultured in MEM supplemented with 10% FBS and incubated at 37 °C in a humidified 5% CO_2_ atmosphere for a total of 96 h.

### 2.8. Treatments with C. lumia Polyphenols-Rich Extract

In the 2D cultures, HepG2 cells were exposed to ClumWp dissolved in the culture medium at concentrations of 5, 15, 25, 50, and 100 µg/mL for 24 h. Conversely, HepG2 + LX-2 spheroids were treated with the concentrated extract, dissolved in the culture medium, at concentrations of 25, 50, and 100 µg/mL for 48 h after seeding. All cell cultures were confirmed to be free of *Mycoplasma* spp. contamination using a specific detection kit (G238, Applied Biological Materials, Richmond, BC, V6V 2J5, Canada) [21].

### 2.9. Cell Viability Assay

For 2D cultures, cell viability was determined through the MTT (3-[4,5-dimethylthiazol-2-yl]-2,5 diphenyl tetrazolium bromide) assay. HepG2 cells (1 × 10^4^ cells/well) were seeded in 96-well plate and allowed to adhere overnight. The next day, the cells were treated with ClumWp at concentrations of 5, 10, 15, 25, and 50 μg/mL for 24 h. After the treatment, 0.5 mg/mL of MTT was added to each well, followed by a 4-h incubation at 37 °C. The resulting formazan crystals were solubilized in DMSO, and absorbance was measured at 570 nm using a microplate reader (Multiskan GO, Thermo Scientific, Denver, CO, USA). Cell viability was calculated relative to untreated controls, which were considered 100% viable.

For HepG2 + LX-2 spheroids, total intracellular adenosine triphosphate (ATP) levels were measured using the Cell-Titer-Glo^®^ 3D cell viability assay (Promega, Madison, WI, USA) according to the manufacturer’s protocol [21]. Single spheroids were transferred to a white 96-well assay plate (Corning, New York, NY, USA) containing 50 µL of PBS. An equal volume (50 µL) of assay reagent was then added to each well, and the mixture was shaken vigorously to ensure reagent penetration. The plate was incubated in the dark at room temperature for 20 min before luminescence was measured using the GloMax^®^ Discover Microplate Reader (Promega), with data acquisition performed using the GloMax^®^ Discover software (v. 3.2.3).

### 2.10. Quantification of Intracellular Neutral Lipid Content

To assess intracellular lipid accumulation, HepG2 cells were seeded on coverslips at a density of 5 × 10^4^ cells/well in 24-well plates. After treatment, the cells were washed with PBS, and fixed with 2% paraformaldehyde for 5 min. Intracellular lipids were stained with Oil Red O (ORO) solution (Sigma-Aldrich, St. Louis, MO, USA) for 20 min, while nuclei were stained with DAPI (Sigma-Aldrich, St. Louis, MO, USA) for the same period. All staining steps were performed at room temperature with the samples protected from direct light. Images were captured using a Thunder Imager Leica (DM4B) equipped with a 100X oil immersion objective. The ORO-stained area was normalized to the number of DAPI-stained nuclei and quantified using ImageJ (v.1.52h, NIH, Bethesda, Rockville, MA, USA). For HepG2 + LX-2 spheroids, cells were fixed in 10% (*w*/*v*) paraformaldehyde (Sigma-Aldrich) for 2 h, followed by overnight incubation in 20% (*w*/*v*) sucrose in PBS (Lonza, Rome, Italy). Spheroids were washed three times with PBS, embedded in OCT Cryomount (Histolab, Västra Frölunda, Sweden), and sectioned into 8-µm-thickness using a cryostat (Leica, Wetzlar, Germany). Sections were mounted on glass slides and stored at −80 °C for 1 h before performing ORO staining [21]. Nuclei were stained with DAPI, and images were captured using a Leica Thunder Imager (DM4B) at 20X magnification. The ORO-stained area was normalized to the number of DAPI-stained nuclei and quantified using ImageJ (v.1.54k, NIH).

### 2.11. Statistical Analysis

Statistical analyses were performed using GraphPad PRISM (version 10.3.1, GraphPad Software, Inc., La Jolla, CA, USA). Data were expressed as mean ± S.E.M. The Shapiro–Wilk test was carried out to evaluate data normality. For normally distributed data, one-way ANOVA followed by Tukey’s post hoc test was applied, whereas non-normally distributed data were analyzed using the Kruskal–Wallis test with Dunn’s post hoc analysis. Comparisons between two groups were performed using either the unpaired two-tailed Student’s *t*-test or the Mann–Whitney U test, depending on the data distribution. All the experiments were conducted in triplicate. Statistical significance was set at *p* < 0.05.

## 3. Results

### 3.1. Identification and Quantification of Eriocitrin and Hesperidin in C. lumia Risso Peel Extracts

High-performance liquid chromatography (HPLC) analysis confirmed the presence of rutinosidic flavanones, specifically eriocitrin and hesperidin, in all the extracts obtained from the peel of *C. lumia* hesperidia. Before passing through the adsorbent resin, the W100 extract showed the highest content of eriocitrin (14.914 mg/g of peel weight, Figure 2A) and hesperidin (7.4 mg/g of peel weight, Figure 2A) compared to Et50, which highlighted an eriocitrin content of 0.176 mg/g (Figure 2B) and a hesperidin content of 0.1125 mg/g (Figure 2B). Furthermore, the Et80 peel extract contained 0.14 mg/g of eriocitrin and 0.077 mg/g of hesperidin (Figure 2C).

### 3.2. Analysis of Citrus lumia Risso Polyphenols-Rich Extracts

After purification of the polyphenols, all the extracts were analyzed qualitatively and quantitatively by HPLC to determine the presence of eriocitrin and hesperidin. The ClumWp extract contained 3.54% of eriocitrin and 2.05% of hesperidin (Figure 3A), whereas the ClumEt50p extract contained 2.68% of eriocitrin and 1.7% of hesperidin (Figure 3B). Similarly, 2.87% of eriocitrin and 1.59% of hesperidin were retrieved in the ClumEt80p extract (Figure 3C).

### 3.3. Characterization of Flavonoid Content in Citrus lumia Risso Peel Extracts

The nitrite-aluminum colorimetric assay was used to quantify total flavonoids. As shown in Table 1, the polyphenol-rich extracts had the highest flavonoid content compared to the corresponding peel extract before passing through the absorbent resin (*p* < 0.001). Specifically, ClumWp extract had a flavonoid content of 18.355 ± 1.607 mg/mL, ClumEt80p had a content of 17.680 ± 3.352 mg/mL, while the flavonoid content in ClumEt50p was 16.394 ± 1.440 mg/mL.

### 3.4. Evaluation of Radicals Scavenging Activity of Citrus lumia Risso Peel Extracts Through Electron Paramagnetic Spectroscopy (EPR)

EPR analysis revealed the characteristic six-line spectrum of the DPPH radical (∫ = 3356 ± 0.9 a.u., Figure 4 and Table 2). The tested *Citrus lumia* extracts showed a low DPPH radical scavenging capacity, with the Et80 and the W100 extracts showing a scavenging percentage of 0.77% (∫ = 3330 ± 0.7 a.u., Figure 4 and Table 2) and 3.81% (∫ = 3228 ± 1.25 a.u., Figure 4 and Table 2), respectively.

Conversely, after the concentration through the polystyrene absorbent resin column, the two extracts showed a higher DPPH radical scavenging ability, with a scavenging percentage of 63.41% for the ClumEt80p extract (∫ = 1128 ± 1.26 a.u., *p* < 0.001, Figure 4 and Table 2) and 58.70% for the ClumWp extract (∫ = 1386 ± 1.02 a.u., *p* < 0.001, Figure 4 and Table 2). Ascorbic acid, used as a positive control, had a scavenging percentage of 94.07% (∫ = 199 ± 0.87 a.u., *p* < 0.001, Figure 4 and Table 2).

Finally, EPR analysis revealed the typical four-line absorption spectrum of the BMPO-OH adduct (∫ = 107 ± 2.24 a.u., Figure 5 and Table 3). The ClumWp extract showed a better scavenging percentage against OH^•^ (42.99%, ∫ = 61 ± 2.78 a.u., *p* < 0.001, Figure 5 and Table 3), whereas ClumEt80p showed a comparable OH^•^ scavenging capability with a percentage of 40.19% (∫ = 62 ± 2.42 a.u., *p* < 0.001, Figure 5 and Table 3). Ascorbic acid had an OH^•^ scavenging percentage of 55.14% (∫ = 48 ± 1.66 a.u., *p* < 0.001, Figure 5 and Table 3).

### 3.5. Treatment with C. lumia Polyphenols-Rich Extract Does Not Affect Cell Viability and Decreases Intracellular Lipids in HepG2 Cells

To ensure that ClumWp was not toxic, HepG2 cells were incubated with increasing concentrations of the extract (5, 15, 25, and 50 µg/mL) for 24 h, and cell viability was measured by an MTT assay. As shown in Figure 6, the polyphenol-rich extract had no effect on cell viability as compared to the control (i.e., 0 µg/mL).

To evaluate the lipid content in hepatocytes, the immortalized human HepG2 liver cell line was incubated with increasing amounts of ClumWp extract (5, 15, 25, and 50 µg/mL). Cells grown in the medium without extract showed a significant cytoplasmic lipid accumulation, highlighted by strong red fluorescence, which was significantly reduced by ClumWp extract at the concentrations of 25 and 50 μg/mL (*p* < 0.01 vs. 0 µg/mL, Figure 7), as shown by Oil Red O staining.

### 3.6. C. lumia Polyphenols-Rich Extract Decreases Intracellular Lipids in 3D HEPG2/LX2 Spheroids

To further confirm these results in different cell types, we investigated the effect of ClumWp extract in spheroids composed of human hepatoma cells (HepG2) and hepatic stellate cells (LX-2) (Figure 8A). Although increasing concentrations of ClumWp altered the ATP concentration (Figure 8B), particularly at 15 (*p* < 0.05), 25 (*p* < 0.01), 50 (*p* < 0.05), and 100 (*p* < 0.01), the volume of the spheroids remained unchanged after treatment (Figure 8C). Thus, consistent with the results in HepG2 cells, the extract was non-toxic to spheroids as well.

In addition, the treatment with 100 µg/mL ClumWp resulted in a reduction in the intracellular lipid content compared to untreated cells (Figure 9).

## 4. Discussion

*Citrus* fruits are widely recognized as a source of several bioactive compounds that provide consumers with nutritional, antioxidant, and anti-inflammatory properties. Particularly, the yield of secondary metabolites, such as polyphenols, which are mainly responsible for these effects, depends on the extraction and the purification procedures [22]. In the extraction step, the solvent chosen represents a fundamental aspect to obtain polyphenols, as their chemical structure implies a greater hydrophilicity than lipophilicity [23]. Thus, methanol could be considered as the most suitable solvent for the extraction of phenolic compounds [24], although more environmentally friendly, non-toxic solvents, such as ethanol and water, can represent safer alternatives for extracting free polyphenols along with aglycones, glycosides, and oligomers, ensuring a high-quality product [25].

*Citrus* species are the predominant components of the Mediterranean maquis [26] and the most popular species, such as *Citrus aurantium* L., *Citrus limon* ((L.) Osbeck), *Citrus bergamia* Risso & Poit., have been extensively studied due to their potential application in human health; indeed, among the polyphenols, flavonoids and phenolic acids represent the main bioactive compounds [27]. However, despite the great diversity of Mediterranean *Citrus* species, research has mainly focused on the more common plants, often neglecting the rarer species, which could potentially be of great interest to increase their use for agro-industrial and pharmaceutical purposes [28]. Recently, some parts of the hesperidium of *C. lumia* Risso have been partially characterized phytochemically and biologically, revealing the presence of bioactive compounds with a strong antioxidant, anti-cholinesterase, and neuroactive properties [9,29,30,31]. Normally, the most abundant flavonoids in *Citrus* peels are neohesperidin, naringin, and neoeriocitrin [32], as well as hesperidin, nobiletin, sinensetin, and tangeretin [33]. In contrast, the most abundant flavonoid in *C. lumia* peels was eriocitrin, followed by hesperidin [9].

Based on these findings, we confirmed the presence of polyphenols in the peel of *C. lumia* (i.e., flavedo and albedo). Specifically, we carried out the extraction of the polyphenolic fraction using a mixture of ethanol:water in different proportions, as well as 100% water, through a maceration process followed by a passage through an adsorption resin to concentrate the polyphenols and remove the sugars from the solution [17,34]. Consistent with data found by others [9,28], our results showed the presence of the rutinosidic flavanones, eriocitrin and hesperidin, in all the extracts from the peel of *C. lumia* hesperidia, reaching the highest content in the W100 extract (eriocitrin (14.914 mg/g of peel weight) and hesperidin (7.4 mg/g of peel weight)), which increased after the concentration step in the column (ClumWp extract: 3.54% of eriocitrin and 2.05% of hesperidin). The determination of total flavonoid content showed that, beyond eriocitrin and hesperidin, ClumWp had the highest concentration of total flavonoids (18.355 ± 1.607 mg/mL). To verify the antioxidant properties of all the enriched extracts, EPR analysis was performed to determine the diversified scavenging activity against DPPH as well as against hydroxyl radical (^•^OH), detected by the BMPO spin trapping technique [34]. Despite a low DPPH radical scavenging capacity observed for ClumWp extract compared to ClumEt80p, the BMPO spin trapping technique showed that ClumWp extract had the best scavenging percentage against OH^•^ (42.99%, ∫ = 61 ± 2.78 a.u.). Based on these results, ClumWp extract was selected to test biological activity in cellular models.

The antioxidant capacity of the polyphenol-rich extracts can be attributed to their high content of flavonoids, especially flavanones, with eriocitrin and hesperidin being the most predominant compounds. Indeed, both eriocitrin and hesperidin have been shown to contribute significantly to the antioxidant capacity of various *Citrus* fruits or their by-products [10,18,35,36], as they are able to scavenge reactive oxygen and nitrogen species through various mechanisms, such as the direct scavenging of free radicals, inhibition of oxidizing enzymes, and activation of antioxidant enzymes [37].

Notably, in addition to the direct neutralization of free radicals, xanthine oxidase inhibition is a key mechanism in counteracting oxidative stress by *Citrus*-derived flavonoids such as hesperidin and its aglycon [38]. Moreover, activation of Nuclear factor E2-related protein 2 (Nrf-2) and the inhibition of nuclear factor kappa B (NF-kB), which are recognized as transcription factors involved in the redox signaling pathway, can induce the expression of intracellular antioxidant enzymes, such as superoxide dismutase type II, glutathione synthase, and NADPH-quinone oxidase (NQO) (HO-1) [39], and prevent the expression of pro-oxidant genes leading to apoptosis, respectively [10,40,41].

In mammalian cells, the endogenous production of free radicals can originate from physiological mechanisms, such as the process of ATP formation in mitochondria, which implies the generation of superoxide radical because of electron transport through the respiratory chain [14]. Thus, the balance between free radical production and the function of endogenous antioxidants can ensure the maintenance of physiological energy status in cells, even in the presence of detrimental exogenous stimuli [14]. Our results show that none of the concentrations of *C. lumia* extract tested (ClumWp, 5, 15, 25 and 50 µg/mL) were toxic to HepG2 cells, which did not show any change in metabolic activity. On the other hand, in spheroids composed of human hepatoma cells (HepG2) and hepatic stellate cells (LX-2), the ATP concentration increased in a dose-dependent manner at 15 and 25 µg/mL, although at higher concentrations the measured ATP was comparable to that found at 15 µg/mL. These peculiar results may be due to an additional effect of flavonoids on mitochondria beyond the antioxidant mechanisms mentioned above. Recent evidence shows that eriocitrin from *C. limon* can induce the expression of ATP5J and COX4I1, encoding proteins involved in ATP synthesis [42]. COX4I1 encodes for subunit 1 of the cytochrome c oxidase, the last enzyme in the mitochondrial electron transport chain, which is allosterically modulated by ATP levels. Specifically, the allosteric ATP inhibition of phosphorylated component 1 allows the maintenance of a physiological mitochondrial membrane potential (known as the relaxed state) to avoid the overproduction of reactive oxygen species [43]. In this view, the lack of ATP synthesis following exposure to higher concentrations could be justified by this regulatory mechanism.

Several studies have demonstrated that *Citrus*-derived eriocitrin can exert antilipemic effects by enhancing mitochondrial biogenesis [10], which is aimed at maintaining mitochondrial mass and function [44], and fatty acid oxidation [45].

In our setting, we showed that the lipid content in hepatocytes was reduced by higher concentrations of ClumWp extract, both in HepG2 cells (25 and 50 μg/mL) and in spheroids (100 μg/mL), suggesting that the fine regulation of metabolic activity at the mitochondrial level, at higher concentrations, could be associated with the enhancement of β-oxidation. In line with this hypothesis, a recent study showed that an extract enriched in polyphenols led to a reduction in intracellular lipid content in human hepatocytes, possibly due to an increase in beta-oxidation [46]. Furthermore, 10 µM of eriocitrin was able to induce an increase in the mitochondrial mass of HepG2 cells and of the expression of NRF1 and TFAM genes, which are involved in mitochondrial biogenesis, while prolonged exposure to 30 µM of eriocitrin for a longer time was able to prevent lipid accumulation in hepatocytes [40]. It has been shown that the induction of NRF1 by PGC-1α can promote the expression of nuclear and mitochondrion-encoded genes of oxidative metabolism (lipid oxidation, and electron transport complexes), capable of promoting mitochondrial biogenesis and, at the same time, counteracting hepatic steatosis [42]. The stimulatory effect of a *C. limon* peel- polyphenolic extract (29.5% eriocitrin, 0.9% hesperidin, 0.5% narirutin, 0.3% diosmin, and 32.9% unknown polyphenols) on hepatic fatty acid oxidation has also been demonstrated *in vivo*. In particular, the activation of PPARα, which induced the overexpression of fatty acid oxidation-related genes, was attributed to eriocitrin and hesperidin [45,46].

## 5. Conclusions

This study provides evidence that the polyphenolic fraction of *Citrus lumia* peel exerts beneficial effects on lipid metabolism and cellular energy balance. These effects are likely to be mediated by its ability to enhance beta-oxidation, promote fatty acid catabolism, and restore ATP production processes, which are disrupted by lipid accumulation in hepatocytes, a hallmark of MASLD and other metabolic disorders [47,48]. These findings provide new insights into the multiple properties and potential applications of our extract, particularly in the prevention of metabolic disorders and supporting the development of novel nutraceuticals for human health.

Moreover, the study of neglected species, such as *Citrus lumia* Risso, offers a valuable opportunity to valorize Mediterranean biodiversity. The extraction of bioactive compounds from plants remains a key strategy for the study of their biology, paving the way for their application in various fields of research.

## Figures and Tables

**Figure 1 plants-14-01209-f001:**
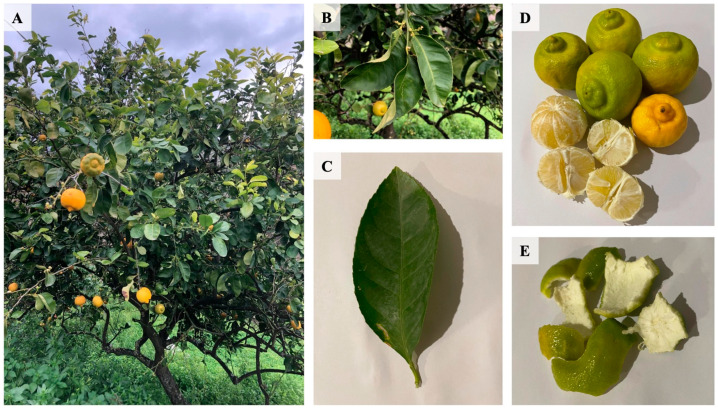
*Citrus lumia* Risso. (**A**) Whole plant, (**B**,**C**) Leaves, (**D**) Hesperidia, (**E**) Albedo and Flavedo.

**Figure 2 plants-14-01209-f002:**
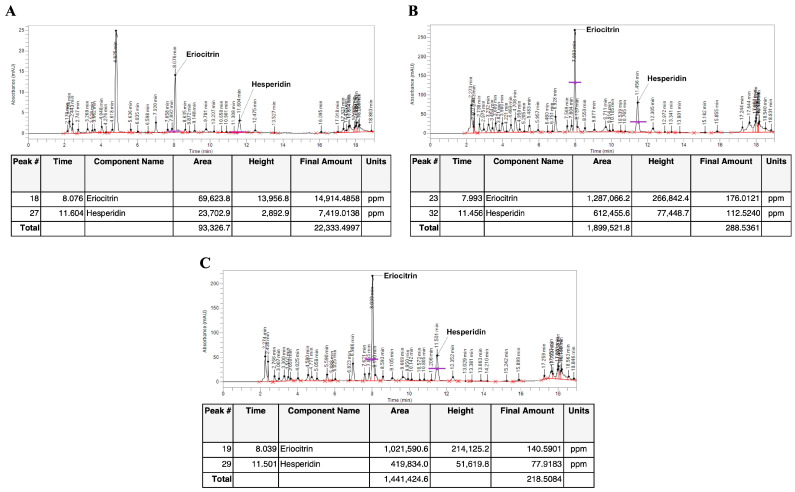
Phytochemical characterization and quantification of *Citrus lumia* Risso peel extracts. HPLC chromatograms of (**A**) W100 (*Citrus lumia* water extract), (**B**) Et50 (*Citrus lumia* 50% EtOH extract), and (**C**) Et80 (*Citrus lumia* 80% EtOH extract).

**Figure 3 plants-14-01209-f003:**
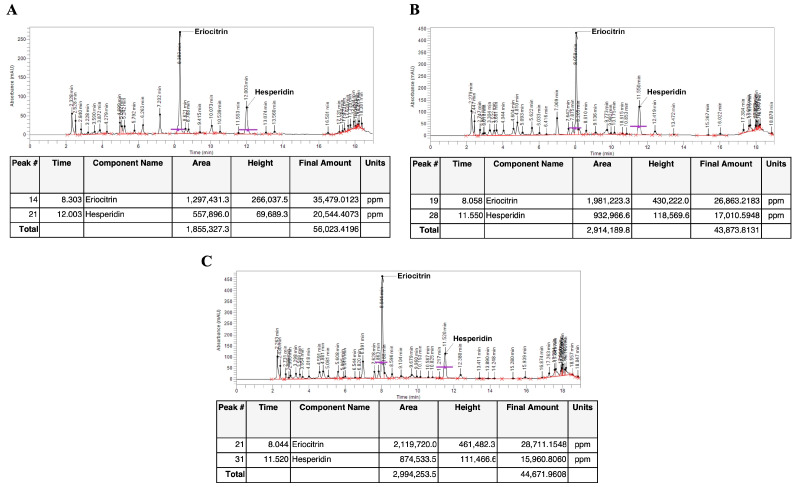
Phytochemical characterization and quantification of *Citrus lumia* Risso polyphenols-rich extracts. HPLC chromatograms of (**A**) ClumWp (*Citrus lumia* water polyphenolic-rich extract), (**B**) ClumEt50p (*Citrus lumia* 50% EtOH polyphenolic-rich extract), and (**C**) ClumEt80p (*Citrus lumia* 80% EtOH polyphenolic-rich extract).

**Figure 4 plants-14-01209-f004:**
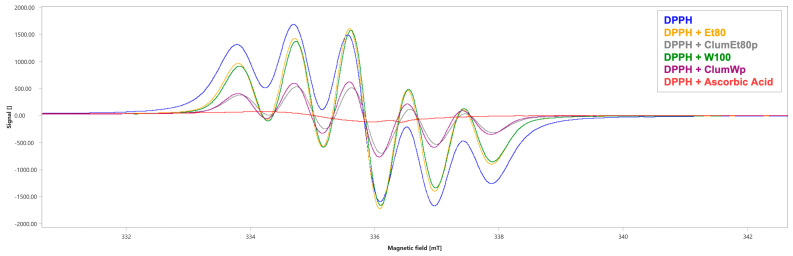
EPR spectra of DPPH in the absence (blue) and presence of the tested extracts. Ascorbic acid was used as a positive control. mT: millitesla. Et80: *Citrus lumia* 80% EtOH extract; W100: *Citrus lumia* water extract; ClumEt80p: *Citrus lumia* 80% EtOH polyphenolic-rich extract; ClumWp: *Citrus lumia* water polyphenolic-rich extract.

**Figure 5 plants-14-01209-f005:**
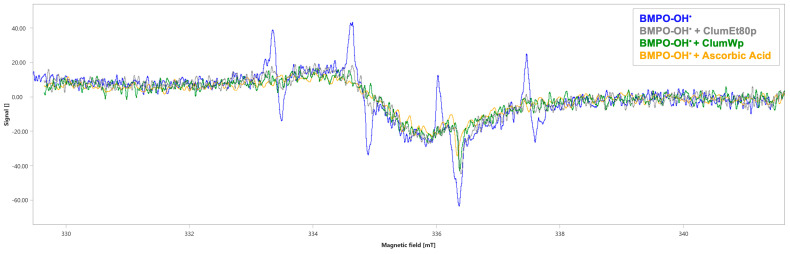
EPR spectra of BMPO-OH^•^ adduct in the absence (blue) and presence of the tested extracts. Ascorbic Acid was used as positive control. mT: millitesla. ClumEt80p: *Citrus lumia* 80% EtOH polyphenolic-rich extract; ClumWp: *Citrus lumia* water polyphenolic-rich extract.

**Figure 6 plants-14-01209-f006:**
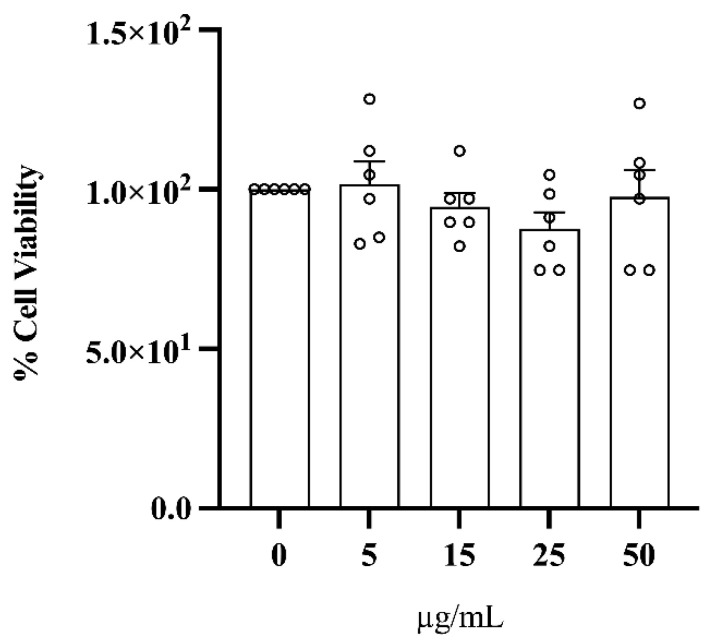
MTT assay for cell viability. Cell viability in response to exposure to increasing concentrations of the ClumWp extract (5, 15, 25, and 50 µg/mL). Results are expressed as mean ± S.E.M.

**Figure 7 plants-14-01209-f007:**
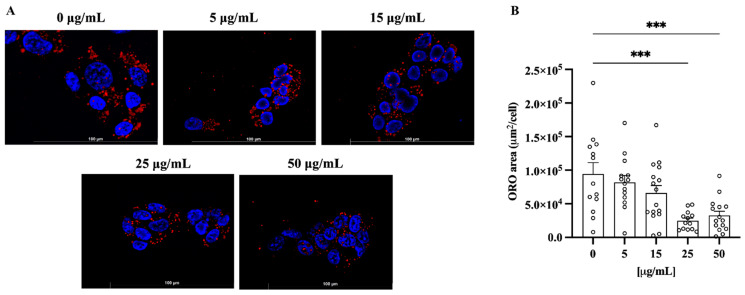
Total fatty acid accumulation in HepG2 cells assessed using Oil Red O (ORO) staining. (**A**) Representative images of lipid accumulation in HepG2 cells. (**B**) Results are expressed as mean ± S.E.M. *** *p* < 0.001 vs. 0 µg/mL.

**Figure 8 plants-14-01209-f008:**
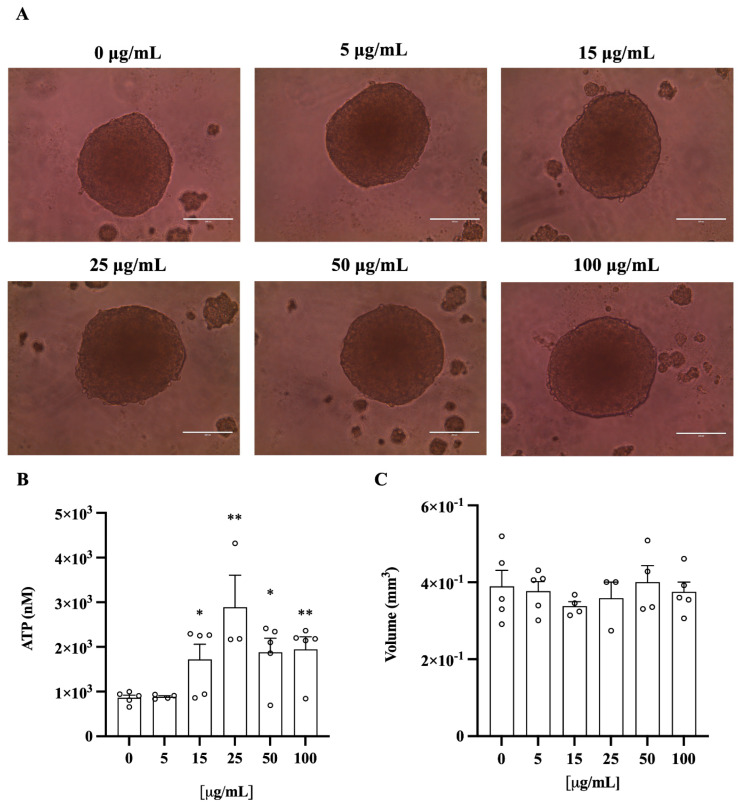
Spheroid volume and viability. (**A**) HepG2 + LX2 cells cultured as 3D spheroids exposed to increasing concentrations of ClumWp (5, 15, 25, 50, and 100 µg/mL). Objective 20X. (**B**) Cellular ATP levels. (**C**) Average volume was calculated by measuring their long and short diameters using the ZEN 2.3 Lite software (Zeiss). Results are expressed as mean ± S.E.M. * *p* < 0.05, ** *p* < 0.01 vs. 0 µg/mL.

**Figure 9 plants-14-01209-f009:**
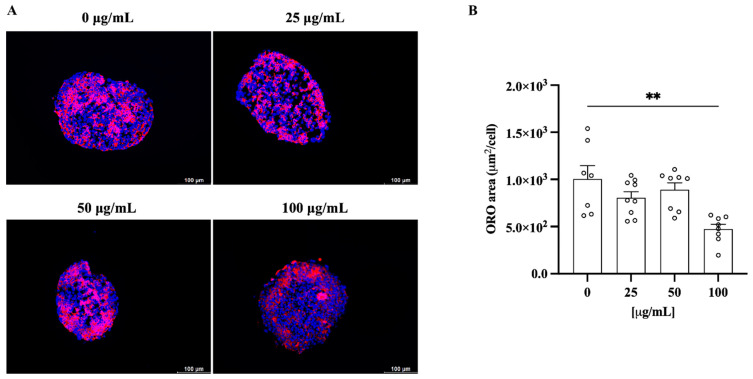
Total fatty acid accumulation in spheroids composed of human hepatoma cells (HepG2) and hepatic stellate cells (LX-2) assessed using Oil Red O (ORO) staining. (**A**) Representative images of lipid accumulation in spheroids in the absence (0 µg/mL) or in the presence of increasing concentrations of ClumWp (25, 50, and 100 µg/mL). (**B**) Results are expressed as mean ± S.E.M. ** *p* < 0.01 vs. 0 µg/mL.

**Table 1 plants-14-01209-t001:** Total Flavonoid Content (TFC) of the *Citrus lumia* extracts.

Sample	ABS_341nm_	TFC (mg/mL)	Mean ± SD (mg/mL)
Et50	0.126	4.216	4.630 ± 0.508
	0.141	5.196	
	0.130	4.477	
Et80	0.115	3.497	3.932 ± 0.399
	0.127	4.281	
	0.123	4.020	
W100	0.118	3.693	4.586 ± 1.173
	0.125	4.150	
	0.152	5.915	
ClumEt50p	0.287	14.739	16.394 ± 1.440 ^a^
	0.323	17.092	
	0.327	17.353	
ClumEt80p	0.273	13.824	17.680 ± 3.352 ^b^
	0.366	19.902	
	0.357	19.314	
ClumWp	0.370	20.163	18.355 ± 1.607 ^c^
	0.323	17.092	
	0.334	17.810	

Abs (Absorbance); S.D. (Standard Deviation). Et50: *Citrus lumia* 50% EtOH extract; Et80: *Citrus lumia* 80% EtOH extract; W100: *Citrus lumia* water extract; ClumEt50p: *Citrus lumia* 50% EtOH polyphenolic-rich extract; ClumEt80p: *Citrus lumia* 80% EtOH polyphenolic-rich extract; ClumWp: *Citrus lumia* water polyphenolic-rich extract. ^a^: *p* < 0.001 vs. Et50; ^b^: *p* < 0.001 vs. Et80; ^c^: *p* < 0.001 vs. W100.

**Table 2 plants-14-01209-t002:** EPR spectroscopy for DPPH radical scavenging activity of *Citrus lumia* extracts.

Sample	Spectral Area (∫, a.u. ± SD)	% Scavenging
DPPH	3356 ± 0.9	
DPPH + Et80	3330 ± 0.7	0.77%
DPPH + W100	3228 ± 1.25	3.81%
DPPH + ClumEt80p	1228 ± 1.26 ^a, b^	63.41%
DPPH + ClumWp	1386 ± 1.02 ^a, c^	58.70%
DPPH + Ascorbic acid	199 ± 0.87 ^a^	94.07%

a.u. (arbitrary units), S.D. (standard deviation). Et80: *Citrus lumia* 80% EtOH extract; W100: *Citrus lumia* water extract; ClumEt80p: *Citrus lumia* 80% EtOH polyphenolic-rich extract; ClumWp: *Citrus lumia* water polyphenolic-rich extract. ^a^: *p* < 0.001 vs. DPPH; ^b^: *p* < 0.001 vs. Et80; ^c^: *p* < 0.001 vs. W100.

**Table 3 plants-14-01209-t003:** EPR spectroscopy for OH^•^ radical scavenging activity of *Citrus lumia* extracts.

Sample	Spectral Area (∫, a.u. ± SD)	% Scavenging
BMPO-OH^•^	107 ± 2.24	
BMPO-OH^•^ + ClumEt80p	64 ± 2.42 ^a^	40.19%
BMPO-OH^•^ + ClumWp	61 ± 2.78 ^a^	42.99%
BMPO-OH^•^ + Ascorbic acid	48 ± 1.66 ^a^	55.14%

a.u. (arbitrary units), S.D. (standard deviation). ClumEt80p: *Citrus lumia* 80% EtOH polyphenolic-rich extract; ClumWp: *Citrus lumia* water polyphenolic-rich extract. ^a^: *p* < 0.001 vs. BMPO-OH^•^.

## Data Availability

The data presented in this study are available on request from the corresponding author.

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
