# Peer review of "Antioxidant and In Vitro Hepatoprotective Activities of a Polyphenol-Rich Fraction from the Peel of Citrus lumia Risso (Rutaceae)"

_plants, 2025, doi:10.3390/plants14081209_

Round 1
Reviewer 1 Report
Comments and Suggestions for Authors
Dear Editor
Many thanks for considering me as a potential reviewer for the article "Phytochemical Analysis, Antioxidant and In vitro Hepatoprotective Activities of a Polyphenol-Rich Fraction from Citrus lumia Risso (Rutaceae)". The article is undoubtedly well-structured, well-presented and well-written. However, I have several observations that should be considered before proceeding further.
Major observations
- The abstract seems to be more descriptive and lacks of findings and recommendations. I will suggest you please re-consider it by incorporating the main goals, method, results and conclusion/recommendation.
- In the section Plant Material; Please add more information about the ecological aspects of the area i.e. weather, humidity, coordinates and collection method. These are the factors mandatory for this kind of study, please consider them.
- I did not see any statement/information, about whether the experiment was conducted in triplicate, duplicate please clearly mention. Another point you had mentioned h and hour, please create consistency.
- ml shoud be mL, please check whole manuscript,
- p < 0.05 should be p < 0.05 and p<0.001 should be p<0.001
- Please improve the quality of figures 2 and 3. Put an arrow over the identified peak, its hard to see, figures captions/description should not be bold, please check whole manuscript
- Table 2 lacks significant letters,
- Line-380 Citrus should be italicized,
- Co conclusion should be improved.
Dear Editor/Authors,
I read the article, its intresting, however, I found several grammatical and technical issues, I will suggest you please have a critical look/may be a minor English editing.
Thank you!!!
Author Response
Comment 1: The abstract seems to be more descriptive and lacks of findings and recommendations. I will suggest you please re-consider it by incorporating the main goals, method, results and conclusion/recommendation.
Response 1: First of all, we would like to thank the reviewer for the constructive criticism and the time spent in analysing this manuscript in depth. We have now incorporated the main aims, methods, results and conclusions accordingly.
Comment 2: In the section Plant Material; Please add more information about the ecological aspects of the area i.e. weather, humidity, coordinates and collection method. These are the factors mandatory for this kind of study, please consider them.
Response 2: We thank the reviewer for this suggestion. Accordingly, we have now improved the Plant Material section by adding the requested information.
Comment 3: I did not see any statement/information, about whether the experiment was conducted in triplicate, duplicate please clearly mention. Another point you had mentioned h and hour, please create consistency.
Response 3: We thank the reviewer for this suggestion. We have now included this statement in the statistical analysis section of the manuscript.
Comment 4: ml should be mL, please check whole manuscript, p < 0.05 should be p < 0.05 and p<0.001 should be p<0.001.
Response 4: We thank the reviewer for this suggestion. We have now made all the requested changes.
Comment 5: Please improve the quality of figures 2 and 3. Put an arrow over the identified peak, it’s hard to see, figures captions/description should not be bold, please check whole manuscript
Response 5: We thank the reviewer for this suggestion. We have now improved Figures 2 and 3 by highlighting the peaks of interest more clearly. Moreover, we have checked all the captions and removed the bold font.
Comment 6: Table 2 lacks significant letters.
Response 6: We thank the reviewer the suggestion. We have now added the significant letters on the data presented in Table 1, 2 and 3 of the manuscript.
Comment 7: Line-380 Citrus should be italicized.
Response 7: We have made the requested change. It was a typo error.
Comment 8: Co conclusion should be improved.
Response 8: We thank the reviewer for this suggestion. We have now improved the conclusions.
Comment 9: I read the article, it’s interesting, however, I found several grammatical and technical issues, I will suggest you please have a critical look/may be a minor English editing.
Response 9: We would like to thank the reviewer; we have now corrected the grammatical and technical problems in the manuscript.
Reviewer 2 Report
Comments and Suggestions for Authors
The paper titled Phytochemical Analysis, Antioxidant and In vitro Hepatoprotective Activities of a Polyphenol-Rich Fraction from Citrus lumia Risso (Rutaceae), besides the content of eriocitrin and hesperidin in various peel extracts, investigated their antioxidant activity using electron paramagnetic resonance (EPR) spectroscopy. Additionally, the extract with the highest flavonoid content and the strongest antioxidant activity against hydroxyl radicals was tested to evaluate its potential protective effects on lipid accumulation in both 2D hepatocyte cultures and 3D spheroids.
In my opinion, the paper could be published after a more thorough revision, in line with the following comments:
According the world flora online accepted name for Citrus lumia Risso is Citrus medica L. (https://www.worldfloraonline.org/search;jsessionid=6DE4E5D3AE5F69E42832EDE63570F66B?query=Citrus+lumia+Risso&limit=24&start=0&sort=)
Introduction
The scientific name of the genus is Citrus L.
According the world flora online (https://www.worldfloraonline.org/taxon/wfo-4000008411) Citrus genus has a wide distribution range, covering many parts of the world, and is particularly known in tropical and subtropical regions, this should be added to the Introduction.
The sentence beginning in line 53 reads: Among the various species, Citrus lumia Risso is an old lime... In my opinion, it should be: Among the various species, Citrus medica L. known as lemon...
Materials and Metodes and Result
Why were only eriocitrin and hesperidin identified and quantified? Other flavonoids and phenolic acids are also strong antioxidants. In my opinion you should also analyzed other compounds in the extracts.
Also, if you only analyzed eriocitrin and hesperidin, why didn’t you determine their activity as well?
Below Table 1, there should be an explanation of the abbreviations in the table.
Figure 2: there should be an explanation of the abbreviations in the text below.
Disscusion
Line 391, the sentence says: Citrus Bergamia, it should be Citrus bergamia.
Author Response
The paper titled Phytochemical Analysis, Antioxidant and In vitro Hepatoprotective Activities of a Polyphenol-Rich Fraction from Citrus lumia Risso (Rutaceae), besides the content of eriocitrin and hesperidin in various peel extracts, investigated their antioxidant activity using electron paramagnetic resonance (EPR) spectroscopy. Additionally, the extract with the highest flavonoid content and the strongest antioxidant activity against hydroxyl radicals was tested to evaluate its potential protective effects on lipid accumulation in both 2D hepatocyte cultures and 3D spheroids.
In my opinion, the paper could be published after a more thorough revision, in line with the following comments:
Comment 1: According to the world flora online accepted name for Citrus lumia Risso is Citrus medica L. (https://www.worldfloraonline.org/search;jsessionid=6DE4E5D3AE5F69E42832EDE63570F66B?query=Citrus+lumia+Risso&limit=24&start=0&sort=)
Response 1: We thank the reviewer for giving us the opportunity to clarify this aspect. Citrus lumia is certainly a critical taxon, described by Risso which assigned formally the binomial name in the following publication “Citrus lumia, Citre lumie. In: Histoire naturelle des principales productions de l’Europe méridionale et particulièrement de celles des environs de Nice et des Alpes Maritimes (A. Risso (1826) 414-418)”.
On a phenetic basis, the fruit of ‘lumia’, as well as the rest of the plant, resembles the lemon (i.e. Citrus limon), from which it differs in its juice, since in C. lumia it is sweet and not sour. The hybrid nature of Citrus lumia is quite clear, as many important systematists of the genus Citrus consider it to be a hybrid of citron and orange (C. medica × C. sinensis), or of citron and pummelo (C. medica L. × C. grandis) or, again, of pummelo and lemon (C. grandis × C. × limon). We believe that there is still much confusion about the identity of C. lumia, largely treated as simple variety of either C. medica or C. limon-whose hybrid nature has since been ascertained- and this in relation to the characters of one or the other species that classical systematists have seen prevailing. However, to date, and to the best of our knowledge, there is a lack of specific molecular-genetic analyses that clearly support these conclusions. Therefore, although the WFO lists Citrus lumia as a synonym for Citrus medica, we prefer to use the binomial C. lumia, as reported before us, by other researchers and eminent systematists.
Comment 2: Introduction The scientific name of the genus is Citrus L.
Response 2: We have modified accordingly.
Comment 3: According to the world flora online (https://www.worldfloraonline.org/taxon/wfo-4000008411) Citrus genus has a wide distribution range, covering many parts of the world, and is particularly known in tropical and subtropical regions, this should be added to the Introduction.
Response 3: We have modified as suggested.
Comment 4: The sentence beginning in line 53 reads: Among the various species, Citrus lumia Risso is an old lime... In my opinion, it should be: Among the various species, Citrus medica L. known as lemon...
Response 4: As explained above, we would prefer to refer to this species as Citrus lumia Risso. Furthermore, stating ‘[..]Citrus medica L. known as lemon’ could be misleading, as Citrus medica is known as citron, while the lemon is Citrus × limon.
Comment 5: Materials and Methods and Result
Why were only eriocitrin and hesperidin identified and quantified? Other flavonoids and phenolic acids are also strong antioxidants. In my opinion you should also analysed other compounds in the extracts.
Also, if you only analysed eriocitrin and hesperidin, why didn’t you determine their activity as well?
Response 5: We thank the reviewer for the comment. The aim of the study was to evaluate the polyphenolic extract titrated in eriocitrin and hesperidin as they are the predominant flavonoids in Citrus lumia. Certainly, we acknowledge that other flavonoids and phenolic acids could also contribute to the overall activity of the extract. Regarding the determination of the biological activity, our primary aim was to evaluate the activity of the whole phytocomplex. While the biological activity of the individual compounds has been well established in previous studies, we recognize that a direct evaluation of eriocitrin and hesperidin within our specific extracts would provide additional insights. We appreciate your suggestion, and, in future, we will consider comparing the biological activity of the individual polyphenols with the biological activity of the polyphenol-enriched phytocomplex extracted from the peel in future research.
Comment 6: Below Table 1, there should be an explanation of the abbreviations in the table.
Figure 2: there should be an explanation of the abbreviations in the text below.
Response 6: We thank the reviewer for this suggestion. We have now improved the legend of all tables and figures by adding an explanation of the missing abbreviations.
Comment 7: Discussion
Line 391, the sentence says: Citrus Bergamia, it should be Citrus bergamia.
Response 7: We have now corrected the sentence.
Reviewer 3 Report
Comments and Suggestions for Authors
The study explores Citrus lumia Risso, a rare Mediterranean lime, focusing on its polyphenol-rich peel extracts. The extracts were characterized, assessed for antioxidant activity, and tested for protective effects against lipid accumulation in hepatocyte models. While the study presents interesting findings, there are some concerns that need to be addressed:
- The current title suggests that the study will discuss different parts of Citrus lumia Risso, whereas the research focuses specifically on the peel. I recommend that the Authors revise the title to more clearly reflect the study's focus on the peel extract.
- The introduction does not provide sufficient discussion on existing literature related to Citrus lumia Risso known biological activities. If there is limited or no existing data on the biological activity highlighting the research gap would emphasize the novelty and significance of the study.
- The Authors emphasize the presence of only two flavonoids in their studies, despite stating that Citrus lumia Risso is characterized by four main flavonoids: eriocitrin, hesperidin, diosmin, and rutin. Additionally, the HPLC chromatograms show more peaks that are not addressed. Could the Authors clarify why only two flavonoids were highlighted in the analysis?
- The Authors claim to have performed an analysis of variance (ANOVA); however, no such results are presented in the tables of the manuscript.
- Could the Authors elaborate on the practical applications of their findings, particularly given that the results are based solely on the peel of Citrus lumia Risso, a species that is extremely difficult to retrieve? Is the proposed approach feasible and economically viable for practical use in food, pharmaceutical, or cosmetic industries?
I recommend that the Authors carefully review the manuscript for language clarity and grammatical accuracy. There are several instances where punctuation and sentence structure need improvement. For example, the sentence:
"Next, they were peeling and, the peel was dried at 40°C for 24 h and then, grinded in liquid nitrogen into a fine powder, with a mortar and pestle, which offering a high extractive surface."
contains multiple language issues, including incorrect verb tense, punctuation errors, and awkward phrasing. There are several similar instances throughout the text that require attention. A thorough revision of the English language, possibly with the assistance of a native speaker or professional language editing service, is strongly recommended to improve clarity and readability.
Author Response
The study explores Citrus lumia Risso, a rare Mediterranean lime, focusing on its polyphenol-rich peel extracts. The extracts were characterized, assessed for antioxidant activity, and tested for protective effects against lipid accumulation in hepatocyte models. While the study presents interesting findings, there are some concerns that need to be addressed:
Comment 1: The current title suggests that the study will discuss different parts of Citrus lumia Risso, whereas the research focuses specifically on the peel. I recommend that the Authors revise the title to more clearly reflect the study's focus on the peel extract.
Response 1: We would like to thank the reviewer for the constructive criticism and the time spent in analysing this manuscript in depth. We have now revised the title to better reflect the focus of the study on the peel extract. The title has been updated as follows: “Phytochemical Analysis, Antioxidant and In vitroHepatoprotective Activities of a Polyphenol-Rich Fraction from the peel of Citrus lumia Risso (Rutaceae)”.
Comment 2: The introduction does not provide sufficient discussion on existing literature related to Citrus lumia Risso known biological activities. If there is limited or no existing data on the biological activity highlighting the research gap would emphasize the novelty and significance of the study.
Response 2: We appreciate the reviewer’s suggestion. As there is no previous evidence showing the beneficial effects of Citrus lumia extracts on lipid accumulation, we have explicitly stated in the Introduction that our aim was to investigate this unexplored aspect and to fill this gap.
Comment 3: The Authors emphasize the presence of only two flavonoids in their studies, despite stating that Citrus lumia Risso is characterized by four main flavonoids: eriocitrin, hesperidin, diosmin, and rutin. Additionally, the HPLC chromatograms show more peaks that are not addressed. Could the Authors clarify why only two flavonoids were highlighted in the analysis?
Response 3: We thank the reviewer for giving us the opportunity to clarify this aspect. Previously, it has been reported that C. lumia have a significant content of the flavanone O-rutinosides hesperidin and eriocitrin. They also have a reduced content of the other flavanone rutinosides narirutin and of the flavones rutin and diosmin. The focus of our study was to evaluate a polyphenolic extract titrated in eriocitrin and hesperidin as they are the most abundant flavonoids in Citrus lumia Risso. Regarding the additional peaks observed in the HPLC chromatograms, we are aware that other flavonoids and phenolic compounds may be present. However, our study aimed to specifically quantify the two major flavonoids due to their relevance. In future analyses, if we have the opportunity, we will quantify additional compounds also using more comprehensive methods. We appreciate your suggestion and will consider this in further research.
Comment 4: The Authors claim to have performed an analysis of variance (ANOVA); however, no such results are presented in the tables of the manuscript.
Response 4: We thank the reviewer for allowing us to better clarify this aspect. The ANOVA followed by Tukey's post hoc test, or the Kruskal-Wallis followed by Dunn’s post hoc test was performed and we have now added the statistical significance in the tables of the results section.
Comment 5: Could the Authors elaborate on the practical applications of their findings, particularly given that the results are based solely on the peel of Citrus lumia Risso, a species that is extremely difficult to retrieve? Is the proposed approach feasible and economically viable for practical use in food, pharmaceutical, or cosmetic industries?
Response 5: We appreciate the reviewer for allowing us to clarify this aspect. Our findings could have several applications:
- As stated in the discussion section, the study of neglected species, such as Citrus lumia Risso, offers a valuable opportunity to valorise Mediterranean biodiversity. Our study extends previous insights on the beneficial effects of this species, which implies the chance to promote its cultivation and the growth of the local economy. This approach has allowed to further valorise other Citrus species, such it occurred for Citrus bergamia, which grows almost exclusively in a small area of Calabria (southern Italy) and has only been used in cosmetic industry in the past.
- The extraction of bioactive compounds from plants remains a key strategy for the study of their biology, paving the way for their application in various fields of research, such as food or pharmaceutical industries. Indeed, as previously demonstrated, this approach permits to highlight the potential of waste by-products such as peels, which are often discarded, thus reducing their negative impact on the environment and promoting their reuse for valuable bioactive compounds.
- Finally, the extraction procedure and the method of concentrating the polyphenolic fraction using adsorbent resins can be applied to other Citrus species, making the approach adaptable and scalable to industrial purposes. This method can promote the use of the extracts as ingredients for the development of nutraceuticals or food supplements aimed at preventing metabolic disorders.
Comment 6: I recommend that the Authors carefully review the manuscript for language clarity and grammatical accuracy. There are several instances where punctuation and sentence structure need improvement. For example, the sentence:
"Next, they were peeling and, the peel was dried at 40°C for 24 h and then, grinded in liquid nitrogen into a fine powder, with a mortar and pestle, which offering a high extractive surface." contains multiple language issues, including incorrect verb tense, punctuation errors, and awkward phrasing. There are several similar instances throughout the text that require attention. A thorough revision of the English language, possibly with the assistance of a native speaker or professional language editing service, is strongly recommended to improve clarity and readability.
Response 6: We would like to thank the reviewer; we have now improved the linguistic clarity and grammatical accuracy of the manuscript.
Reviewer 4 Report
Comments and Suggestions for Authors
The submitted manuscript is well-structured and exhibits a high level of scientific soundness, also contributing a degree of novelty since the studied species, Citrus lumia Risso, has received much less attention in the scientific literature compared to other more common and commercially important citrus species, such as Citrus sinensis, Citrus limon, or Citrus reticulata. That being said, I have noted a lack of fluidity in the text, and I have observed some inconsistency, for example, in the use of units of measurement, spaces, or superscripts. I believe that not paying enough attention to these details negatively impacts the overall perception of the article's quality, and I would recommend that the authors review these aspects.
The objectives are very well reasoned and formulated, and the experimental section explains, in great detail, how each of the assays was performed. The results are expressed clearly, although in my opinion, some figures lack sufficient size and resolution and should be improved, particularly the chromatograms. Finally, the presented results and scientific literature effectively support the discussion, adding significant depth to the authors' reasoning and conclusions.
Given the above, I believe that the submitted manuscript perfectly fits the "Scope & Aims" of Plants, in addition to meeting the criteria of scientific rigor and novelty required in a publication of these characteristics. Therefore, I would recommend its publication after revision by the authors.
Below, I share with the authors other specific comments:
Comment 1: Keywords. There are too many keywords (10); I would suggest the authors reduce the number and eliminate redundancy.
Comment 2: 1. Introduction. Lines 71-75. I recommend the authors review the grammar and revise these lines.
Comment 3: 1. Introduction. Line 84. “ …the imbalanced energy status, free radical over production and…” "Overproduction" should be written as one word, without a space.
Comment 4: 2.2. Extraction procedure. a) Please standardize the spacing between the numerical value and the degree Celsius symbol (°C) throughout the manuscript. b) Why did the authors not evaluate the extraction yields for eriocitrin and hesperidin using a solvent more suitable for their complete recovery (such as methanol and DMSO combinations, or even alkaline solutions), and/or extraction methods more effective for this purpose, such as ultrasonic-assisted extraction? This would provide the authors with insights into the efficacy of their maceration protocol in terms of bioactive compound recovery.
Comment 5: 2.3. Preparation of Citrus lumia Extracts Concentrated in Polyphenols. Hesperidin is practically insoluble in water and poorly soluble in ethanol (https://doi.org/10.1021/je500206w, J. Chem. Eng. Data 2014, 59, 2065−2069), which could compromise its recovery in various stages of the process described by the authors, including extraction and the polyphenol enrichment phase using adsorbent resins.
For example, in this section, the authors describe that the crude extracts “...were diluted with distilled water to a total volume of 200 ml and then passed through a glass column containing 100 ml of adsorbent resins...” This dilution in water, depending on the extract concentration, could cause hesperidin to precipitate before contacting the resin, and perhaps elution with 50% ethanol may not be sufficiently effective to re-dissolve or efficiently elute the hesperidin.
My first question in this regard would be whether the authors have evaluated the recovery of eriocitrin and hesperidin in the concentrated extracts (i.e., ClumWp, ClumEt50p, and ClumEt80p) compared to the crude extracts (i.e., W100m, Et50, and Et80).
My second concern is why the authors did not employ alkaline elution, either aqueous or hydroalcoholic, to ensure the most effective elution of the two selected bioactive compounds.
In this context, when the authors mention the selected resin, they cite reference [14] (line 136); however, that publication does not provide a detailed description or explanation of the resin selection, leading to another publication (Nutrients 2022, 14, 477. https://doi.org/10.3390/nu14030477), in which polyphenol elution is performed in an alkaline medium with potassium hydroxide.
Comment 6: 2.5. Total Flavonoid Content. Line 164. “Absorbance was recorded immediately at the peak wavelength (λ) between 200 and 600 nm.” Please specify the wavelength used to obtain the absorbance and construct the calibration curves. In the referenced article (i.e.,[15]), the following is stated:
“Table 3. Quantification of flavonoids in citrus extracts according to the traditional nitrite-aluminum assay (λ~510 nm) and comparison with the new approach (λ~375 nm). Dilution Factor (DF) was 1:50 in all samples.”
If the authors used the wavelength of the new approach described in [15], I would recommend including this information in the text for better clarity and comprehension.
Comment 7: 2.6 Evaluation of Radicals Scavenging by Electron Paramagnetic Spectroscopy (EPR). Line 171. The acronym DPPH is introduced in the manuscript without prior definition. To improve readability and comprehension for those unfamiliar with this technique, I recommend that the authors provide the full name of this compound upon its first mention.
Comment 8: 2.7 2D and 3D Cell Culture. Line 201. “..cultured in in MEM..”. The word "in" is duplicated.
Comment 8: 2.7 2D and 3D Cell Culture. Line 228. “..Cell-Titer-Glo®..”. The ® symbol should be in superscript. Please review the entire manuscript for similar instances.
Comment 9: 3.1. Identification and Quantification of Eriocitrin and Hesperidin in C. lumia Risso peel extracts. Hesperidin is more soluble in ethanol and hydroalcoholic solutions than in water. However, in the extracts obtained before passing through the columns, the concentrations of both flavonoids (eriocitrin and hesperidin) are significantly higher in the aqueous extract than in the ethanolic extracts (50% and 80% ethanol). From a chemical standpoint, this is rather curious. To what do the authors attribute this phenomenon?
Comments on the Quality of English Language
The article could be improved in terms of fluency and consistency. There are also some grammatical errors that need to be corrected before publication.
Author Response
The submitted manuscript is well-structured and exhibits a high level of scientific soundness, also contributing a degree of novelty since the studied species, Citrus lumia Risso, has received much less attention in the scientific literature compared to other more common and commercially important citrus species, such as Citrus sinensis, Citrus limon, or Citrus reticulata. That being said, I have noted a lack of fluidity in the text, and I have observed some inconsistency, for example, in the use of units of measurement, spaces, or superscripts. I believe that not paying enough attention to these details negatively impacts the overall perception of the article's quality, and I would recommend that the authors review these aspects.
The objectives are very well reasoned and formulated, and the experimental section explains, in great detail, how each of the assays was performed. The results are expressed clearly, although in my opinion, some figures lack sufficient size and resolution and should be improved, particularly the chromatograms. Finally, the presented results and scientific literature effectively support the discussion, adding significant depth to the authors' reasoning and conclusions.
Given the above, I believe that the submitted manuscript perfectly fits the "Scope & Aims" of Plants, in addition to meeting the criteria of scientific rigor and novelty required in a publication of these characteristics. Therefore, I would recommend its publication after revision by the authors.
Below, I share with the authors other specific comments:
Comment 1: Keywords. There are too many keywords (10); I would suggest the authors reduce the number and eliminate redundancy.
Response 1: First or all, we would like to thank the reviewer for the constructive criticism and the time spent in analysing the manuscript in depth. We have now reduced the number of keywords from ten to seven.
Comment 2: 1. Introduction. Lines 71-75. I recommend the authors review the grammar and revise these lines.
Response 2: We thank the reviewer for this suggestion. We have checked the grammar and made the necessary changes to these lines.
Comment 3: 1. Introduction. Line 84. “ …the imbalanced energy status, free radical over production and…” "Overproduction" should be written as one word, without a space.
Response 3: We have made the requested change. It was a typo error.
Comment 4: 2.2. Extraction procedure. a) Please standardize the spacing between the numerical value and the degree Celsius symbol (°C) throughout the manuscript. b) Why did the authors not evaluate the extraction yields for eriocitrin and hesperidin using a solvent more suitable for their complete recovery (such as methanol and DMSO combinations, or even alkaline solutions), and/or extraction methods more effective for this purpose, such as ultrasonic-assisted extraction? This would provide the authors with insights into the efficacy of their maceration protocol in terms of bioactive compound recovery.
Response 4: We thank the reviewer for this suggestion and for the opportunity to better clarify the reason for the chosen extraction procedure. We have now standardised the spacing between the number and the degree Celsius symbol throughout the manuscript.
Furthermore, we agree that organic solvents, such as methanol and DMSO combinations or alkaline solutions, as well as advanced extraction techniques, such as ultrasound-assisted extraction, could potentially improve the recovery of eriocitrin and hesperidin. However, despite the advantages of these solvents, we have chosen to use a green chemistry technique, which is characterised by minimal environmental impact and high health safety. As previously described, the next step will be to optimise the recovery of bioactive compounds by coupling maceration with ultrasound sonication, while maintaining our green approach.
Comment 5: 2.3. Preparation of Citrus lumia Extracts Concentrated in Polyphenols. Hesperidin is practically insoluble in water and poorly soluble in ethanol (https://doi.org/10.1021/je500206w, J. Chem. Eng. Data 2014, 59, 2065−2069), which could compromise its recovery in various stages of the process described by the authors, including extraction and the polyphenol enrichment phase using adsorbent resins.
For example, in this section, the authors describe that the crude extracts “...were diluted with distilled water to a total volume of 200 ml and then passed through a glass column containing 100 ml of adsorbent resins...” This dilution in water, depending on the extract concentration, could cause hesperidin to precipitate before contacting the resin, and perhaps elution with 50% ethanol may not be sufficiently effective to re-dissolve or efficiently elute the hesperidin.
My first question in this regard would be whether the authors have evaluated the recovery of eriocitrin and hesperidin in the concentrated extracts (i.e., ClumWp, ClumEt50p, and ClumEt80p) compared to the crude extracts (i.e., W100m, Et50, and Et80).
My second concern is why the authors did not employ alkaline elution, either aqueous or hydroalcoholic, to ensure the most effective elution of the two selected bioactive compounds.
In this context, when the authors mention the selected resin, they cite reference [14] (line 136); however, that publication does not provide a detailed description or explanation of the resin selection, leading to another publication (Nutrients 2022, 14, 477. https://doi.org/10.3390/nu14030477), in which polyphenol elution is performed in an alkaline medium with potassium hydroxide.
Response 5: We thank the reviewer for allowing us to better clarify this aspect.
Regarding the first concern, we performed HPLC analysis on both crude extracts and extracts after the passage through the adsorbent resins. The analysis showed a higher content of eriocitrin and hesperidin in the polyphenols-rich extracts, as shown in Figure 3. Together with the HPLC analysis, the characterisation of the total flavonoid content confirmed that the concentrated extracts had a higher flavonoid content rather than the corresponding crude extract.
Regarding the second concern, we confirmed that the elution of the polyphenolic fraction was carried out using a mild KOH solution. The detailed procedure was described for the first time in “Mollace, V., Sacco, I., Janda, E., Malara, C., Ventrice, D., Colica, C., Visalli, V., Muscoli, S., Ragusa, S., Muscoli, C., Rotiroti, D., & Romeo, F. (2011). Hypolipemic and hypoglycaemic activity of bergamot polyphenols: from animal models to human studies. Fitoterapia, 82(3), 309–316. https://doi.org/10.1016/j.fitote.2010.10.014” and this standardised method was used for all the successive studies to concentrate polyphenols from plants.
Comment 6: 2.5. Total Flavonoid Content. Line 164. “Absorbance was recorded immediately at the peak wavelength (λ) between 200 and 600 nm.” Please specify the wavelength used to obtain the absorbance and construct the calibration curves. In the referenced article (i.e.,[15]), the following is stated:
“Table 3. Quantification of flavonoids in citrus extracts according to the traditional nitrite-aluminum assay (λ~510 nm) and comparison with the new approach (λ~375 nm). Dilution Factor (DF) was 1:50 in all samples.”
If the authors used the wavelength of the new approach described in [15], I would recommend including this information in the text for better clarity and comprehension.
Response 6: We thank the reviewer for this suggestion. We have now specified the wavelength used.
Comment 7: 2.6 Evaluation of Radicals Scavenging by Electron Paramagnetic Spectroscopy (EPR). Line 171. The acronym DPPH is introduced in the manuscript without prior definition. To improve readability and comprehension for those unfamiliar with this technique, I recommend that the authors provide the full name of this compound upon its first mention.
Response 7: We thank the reviewer for this suggestion. We have now included the full name of the compound when it is first mentioned.
Comment 8: 2.7 2D and 3D Cell Culture. Line 201. “..cultured in in MEM..”. The word "in" is duplicated. Line 228. “..Cell-Titer-Glo®..”. The ® symbol should be in superscript. Please review the entire manuscript for similar instances.
Response 8: We have made the requested change, and we have now added a superscript to each ® symbol in the manuscript.
Comment 9: 3.1. Identification and Quantification of Eriocitrin and Hesperidin in C. lumia Risso peel extracts. Hesperidin is more soluble in ethanol and hydroalcoholic solutions than in water. However, in the extracts obtained before passing through the columns, the concentrations of both flavonoids (eriocitrin and hesperidin) are significantly higher in the aqueous extract than in the ethanolic extracts (50% and 80% ethanol). From a chemical standpoint, this is rather curious. To what do the authors attribute this phenomenon?
Response 9: We thank the reviewer for allowing us to better clarify this aspect. Hesperidin is a glycosylated flavonoid, and its solubility could be influenced by the presence of sugar moieties. While aglycones generally exhibit higher solubility in ethanol, glycosylated flavonoids, such as hesperidin, may have better solubility in water due to increased hydrophilicity.
Another study examined a green extraction of polyphenols from citrus peel and confirmed that higher levels of hesperidin were found in the water extract than in ethanolic or hydroalcoholic extracts. The authors reported that hesperidin has more hydrophilic characteristic than other phenolics [Liu, Y., Benohoud, M., Yamdeu, J. H. G., Gong, Y. Y., & Orfila, C. (2021). Green extraction of polyphenols from citrus peel by-products and their antifungal activity against Aspergillus flavus. Food Chemistry: X, 12, 100144.].
Finally, this phenomenon is due to the high hydrophilicity of the hesperidin.
Round 2
Reviewer 1 Report
Comments and Suggestions for Authors
Dear Editor
Many thanks for the updates regarding the article status. I am pleased that the authors have taken into consideration my observations. Anyhow, below are a few minor suggestions that should be addressed,
- In the main theme ‘In vitro’ should be italicized,
- Please consider abstract a single paragraph,
- Figures 4 and 5: Please keep the legends of the figure inside and/or add boundary lines,
- Material and methods, a few protocols are not cited please cite them accordingly,
- Conclusion is now interesting and can be a separate section like others.
Thank you!!
Comments on the Quality of English LanguageDear Editor/Authors,
Thank you for accepting and incorporating my suggestions in your article.
Now it's much more refined, however, a few technical points should be taken into consideration.
Please have a look at the English language still needs technical improvements.
Thank you!
Author Response
Many thanks for the updates regarding the article status. I am pleased that the authors have taken into consideration my observations. Anyhow, below are a few minor suggestions that should be addressed,
Comment 1: In the main theme ‘In vitro’ should be italicized
Response 1: We thank the reviewer for this suggestion. We have now italicised “in vitro” in the main theme.
Comment 2: Please consider abstract a single paragraph,
Response 2: The abstract is already presented as a single paragraph.
Comment 3: Figures 4 and 5: Please keep the legends of the figure inside and/or add boundary lines,
Response 3: We thank the reviewer for this suggestion. The legends in Figures 4 and 5 are now boundary lines and within the figure.
Comment 4: Material and methods, a few protocols are not cited please cite them accordingly
Response 4: We thank the reviewer for this suggestion. We have now cited all the protocols accordingly.
Comment 5: Conclusion is now interesting and can be a separate section like others.
Response 5: The conclusion is now presented as a separate section.
Comment 6: Dear Editor/Authors, thank you for accepting and incorporating my suggestions in your article. Now it's much more refined, however, a few technical points should be taken into consideration. Please have a look at the English language still needs technical improvements.
Response 6: We thank the reviewer for this suggestion. We will now look at the language and improve it where necessary.
Reviewer 2 Report
Comments and Suggestions for Authors
Considering the author's response to my suggestions, the Introduction must be supplemented. In the introduction, the authors should provide the source of the literature they used for the taxon name, but also mention that according to the new nomenclature sources (WFO), it is Citrus medica L.
Also, for a reputable journal like Plants, I believe it is necessary to perform a detailed chemical analysis of the extract, not just the determination of eriocitrin and hesperidin.
Author Response
Comment 1: Considering the author's response to my suggestions, the Introduction must be supplemented. In the introduction, the authors should provide the source of the literature they used for the taxon name, but also mention that according to the new nomenclature sources (WFO), it is Citrus medica L.
Response 1: We have now included the sources we used for the taxon name in the introduction. In addition, the new nomenclature sources (WFO) have been added accordingly.
Comment 2: Also, for a reputable journal like Plants, I believe it is necessary to perform a detailed chemical analysis of the extract, not just the determination of eriocitrin and hesperidin.
Response 2: We thank the reviewer for the suggestion. Since the phytochemical profile of Citrus lumia Risso was previously described in “Smeriglio, A., Alloisio, S., Raimondo, F. M., Denaro, M., Xiao, J., Cornara, L., & Trombetta, D. (2018). Essential oil of Citrus lumia Risso: Phytochemical profile, antioxidant properties and activity on the central nervous system. Food and Chemical Toxicology, 119, 407-416.” and in “Smeriglio, A., Denaro, M., Di Gristina, E., Mastracci, L., Grillo, F., Cornara, L., & Trombetta, D. (2022). Pharmacognostic approach to evaluate the micromorphological, phytochemical and biological features of Citrus lumia seeds. Food chemistry, 375, 131855. https://doi.org/10.1016/j.foodchem.2021.131855”, in our study we only focused on the quantification of eriocitrin and hesperidin, as phytochemical markers for Citrus lumia Risso as well as the most predominant flavonoids in this species. We acknowledge that quantifying additional compounds observed in the chromatograms would provide valuable insights, as they contribute to the overall activity of the extract. However, we are unable to perform further chemical analyses currently. Nonetheless, we recognize the importance of a more comprehensive characterization and consider this an interesting point to address in future research.
Reviewer 3 Report
Comments and Suggestions for Authors
The revised manuscript shows improvement in clarity, structure, and presentation. However, some of the comments were only partially addressed and require further attention.
Comment 2: Except for the hepatoprotective activities, the Authors performed antioxidant assays; however, no comparative discussion with existing literature is provided. Several studies have previously investigated the antioxidant properties of citrus-derived polyphenols—particularly hesperidin and eriocitrin—in similar experimental settings.
Comment 3: The revised manuscript still emphasizes only two flavonoids (hesperidin and eriocitrin), despite the Authors’ own acknowledgment of additional compounds visible in the chromatograms. While the justification for quantifying only the major flavonoids is noted, it does not fully align with the broad claim of a "phytochemical analysis" in the title.
Author Response
Comment 1: The revised manuscript shows improvement in clarity, structure, and presentation. However, some of the comments were only partially addressed and require further attention. Except for the hepatoprotective activities, the Authors performed antioxidant assays; however, no comparative discussion with existing literature is provided. Several studies have previously investigated the antioxidant properties of citrus-derived polyphenols—particularly hesperidin and eriocitrin—in similar experimental settings.
Response 1: We thank the reviewer for this suggestion. Regarding the antioxidant activities of eriocitrin and hesperidin, we have now discussed this aspect by citing the existing literature to better contextualise our findings.
Comment 2: The revised manuscript still emphasizes only two flavonoids (hesperidin and eriocitrin), despite the Authors’ own acknowledgment of additional compounds visible in the chromatograms. While the justification for quantifying only the major flavonoids is noted, it does not fully align with the broad claim of a "phytochemical analysis" in the title.
Response 2: As described in Round 1 of the revision, the focus of our study was to quantify eriocitrin and hesperidin as chemical markers for Citrus lumia. Furthermore, as already reported by others, they are the most abundant flavonoids in Citrus lumia Risso. Moreover, the phytochemical profile of Citrus lumia Risso was already previously described in “Smeriglio, A., Denaro, M., Di Gristina, E., Mastracci, L., Grillo, F., Cornara, L., & Trombetta, D. (2022). Pharmacognostic approach to evaluate the micromorphological, phytochemical and biological features of Citrus lumia seeds. Food chemistry, 375, 131855. https://doi.org/10.1016/j.foodchem.2021.131855”.
We are aware and agree with the reviewer that it would be interesting to quantify other compounds observed in the chromatograms. However, as we do not have the possibility to perform additional HPLC analyses on our extracts, we will modify the title by removing "phytochemical analysis" to better reflect our research, as suggested. The new proposed title will be “Antioxidant and In Vitro Hepatoprotective Activities of a Polyphenol-Rich Fraction from the Peel of Citrus lumia Risso (Rutaceae).”
Reviewer 4 Report
Comments and Suggestions for Authors
The authors have implemented the necessary changes to improve the robustness and consistency of the article. In addition, they have responded objectively and with the support of scientific literature to the methodological doubts raised during the first revision. Finally, they have also significantly improved the English and the fluency of the text.
For future publications, I would recommend that the authors provide a more detailed analysis of the recoveries of target bioactive compounds from their extracts. This would facilitate a robust comparison of the different solvents used, and a thorough evaluation of the extraction methodologies employed.
Without further ado, and after having reviewed the changes made by the authors, I would recommend the publication of the article as it is in this revised version.
Author Response
Comment 1: The authors have implemented the necessary changes to improve the robustness and consistency of the article. In addition, they have responded objectively and with the support of scientific literature to the methodological doubts raised during the first revision. Finally, they have also significantly improved the English and the fluency of the text. For future publications, I would recommend that the authors provide a more detailed analysis of the recoveries of target bioactive compounds from their extracts. This would facilitate a robust comparison of the different solvents used, and a thorough evaluation of the extraction methodologies employed. Without further ado, and after having reviewed the changes made by the authors, I would recommend the publication of the article as it is in this revised version.
Response 1: We sincerely appreciate the positive evaluation of our revisions and the recognition of the improvements in robustness, consistency and clarity of our article. We also appreciate the suggestion to include a more detailed analysis of the recoveries of target bioactive compounds in future publications. We thank the reviewer for the time and effort in reviewing our manuscript.